# Volunteer based approach to dog vaccination campaigns to eliminate human rabies: Lessons from Laikipia County, Kenya

Adam W. Ferguson[1,¤,☯]*, Dishon Muloi[2,3,4☯], Dedan K. Ngatia[5☯], Wangechi Kiongo[5], Duncan M. Kimuyu[5], Paul W. Webala[6], Moses O. Olum[7], Mathew Muturi[8], Samuel M. Thumbi[9,10], Rosie Woodroffe[11], Lucy Murugi[12], Eric M. Fèvre[4,13], Suzan Murray[1], Dino J. Martins[14,15]

1 Global Health Program, Smithsonian Conservation Biology Institute, Washington, United States of America, 2 Usher Institute of Population Health Sciences & Informatics, University of Edinburgh, Charlotte Auerbach Road,Edinburgh, United Kingdom, 3 Centre for Immunity, Infection and Evolution, University of Edinburgh, Edinburgh, United Kingdom, 4 International Livestock Research Institute,Nairobi, Kenya, 5 School of Natural Resources & Environmental Studies, Karatina University, Karatina, Kenya, 6 Department of Forestry and Wildlife Management, Maasai Mara University, Narok, Kenya, 7 Kenya Agricultural and Livestock Research Organization, Muguga North,Kikuyu, Kenya, 8 Kenya Zoonotic Disease Unit, Ministry of Health and Ministry of Agriculture, Livestock, and Fisheries, Nairobi, Kenya, 9 Paul G. Allen School for Global Animal Health, Washington State University, Pullman, Washington, United States of America, 10 Rabies Free Africa, Washington State University, Pullman, Washington, United States of America, 11 Institute of Zoology, Zoological Society of London, Regent's Park, London, United Kingdom, 12 Ministry of Agriculture, Livestock, and Fisheries, County Government of Laikipia, Nanyuki, Kenya, 13 Institute of Infection, Veterinary & Ecological Sciences, University of Liverpool, Leahurst Campus, Neston, United Kingdom, 14 Mpala Research Centre, Nanyuki, Kenya, 15 Department of Ecology and Evolution, Princeton University, Princeton, New Jersey, United States of America

☯ These authors contributed equally to this work.
¤ Current address: Gantz Family Collection Center, Field Museum of Natural History, South Lake Shore Drive, Chicago, Illinois United States of America
* adamwferguson@gmail.com

**Data Availability Statement:** All relevant data are within the manuscript and its Supporting Information files.

## Abstract

### Background

An estimated 59,000 people die from rabies annually, with 99% of those deaths attributable to bites from domestic dogs (*Canis lupus familiaris*). This preventable Neglected Tropical Disease has a large impact across continental Africa, especially for rural populations living in close contact with livestock and wildlife. Mass vaccinations of domestic dogs are effective at eliminating rabies but require large amounts of resources, planning, and political will to implement. Grassroots campaigns provide an alternative method to successful implementation of rabies control but remain understudied in their effectiveness to eliminate the disease from larger regions.

### Methodology/Principal Findings

We report on the development, implementation, and effectiveness of a grassroots mass dog rabies vaccination campaign in Kenya, the Laikipia Rabies Vaccination Campaign. During 2015–2017, a total of 13,155 domestic dogs were vaccinated against rabies in 17 communities covering approximately 1500 km². Based on an estimated population size of 34,275

**Funding:** Sources of funding that supported this work included a National Science Foundation Postdoctoral Fellowship in Biology FY 2014, Award No. DBI-1402456 (AWF, PWW) https://www.nsf.gov/ and a Smithsonian Mpala Postdoctoral George E. Burch Fellow at the National Zoological Park, Smithsonian Institution (AWF, DJM, SM) https://www.smithsonianofi.com/fellowship-opportunities/mpala-postdoctoral-fellowship/. DKN was supported through the National Geographic Society and the Rufford Foundation. SM Thumbi receives funding support from the Wellcome Trust (Grant numbers 110330/Z/15/Z). Some personnel (EMF, DM, other ILRI staff) involved with this study were supported by the UK Medical Research Council, Biotechnology and Biological Science Research Council (UK), the Economic and Social Research Council (UK), the Natural Environment Research Council (UK), through the Environmental & Social Ecology of Human Infectious Diseases Initiative (ESEI), Grant Reference: G1100783/1. These staff were also supported in part by the CGIAR Research Program on Agriculture for Nutrition and Health (A4NH), led by the International Food Policy Research Institute (IFPRI); we acknowledge the CGIAR Fund Donors (https://www.cgiar.org/funders/). Financial support was also provided by Mpala Wildlife Foundation, Laikipia Ranches and Conservation Community and Conservancies, Veterinarians International, D. and C. Keller, Bruce L., and G. and J. Wintroub. The funders had no role in study design, data collection and analysis, decision to publish, or preparation of the manuscript.

**Competing interests:** I have read the journal's policy and the authors of this manuscript have the following competing interests: membership in a government or other advisory board related to rabies elimination (DKN, MM, SMT, LM, and EMF) and relationships with organizations and funding bodies including nongovernmental organizations, research institutions, or charities (AWF, DKN, WK, DK, SM, and DJM).

domestic dogs, percent coverages increased across years, from 2% in 2015 to 24% in 2017, with only 3 of 38 community-years of vaccination exceeding the 70% target. The average cost of vaccinating an animal was $3.44 USD *with* in-kind contributions and $7.44 USD *without* in-kind contributions.

## Conclusions/Significance

The evolution of the Laikipia Rabies Vaccination Campaign from a localized volunteer-effort to a large-scale program attempting to eliminate rabies at the landscape scale provides a unique opportunity to examine successes, failures, and challenges facing grassroots campaigns. Success, in the form of vaccinating more dogs across the study area, was relatively straightforward to achieve. However, lack of effective post-vaccination monitoring and education programs, limited funding, and working in diverse community types appeared to hinder achievement of 70% coverage levels. These results indicate that grassroots campaigns will inevitably be faced with a philosophical question regarding the value of local impacts versus their contributions to a larger effort to eliminate rabies at the regional, country, or global scale.

## Author summary

Given the importance of mass vaccinations of domestic dogs towards eliminating human rabies in Africa and the site-specific challenges facing such campaigns, additional studies on the development and implementation of such efforts are needed. One mechanism of mass vaccination lies in grassroots efforts that often begin at a very local scale and either develop into larger campaigns, remain local, or cease to persist past several years once interest and funding is exhausted. Here, we discuss the development of a grassroots campaign in Laikipia County, Kenya from its local inception to its development into a county-wide rabies elimination effort. Our results highlight challenges associated with achieving the targeted 70% coverage rate, including a need for consistent and systematic demographic monitoring of dog populations, limitations of the central point method, and logistical and financial challenges facing a volunteer-based effort. Serious political commitment from both the local and national governments are necessary to take the budget beyond what a crowdfunded campaign can raise, including availability and access to quality dog rabies vaccines. Without such outside support and substantial time to grow, grassroots campaigns might be better relegated to raising awareness and vaccinating dogs in small communities to protect those communities directly, without contributing to the broader ecosystem-wide transmission-stopping aim often sought by government human health and veterinary organizations.

## Introduction

Rabies represents one of the most important viral Neglected Tropical Diseases (NTDs) in sub-Saharan Africa [1, 2]. This deadly (albeit preventable) disease is characterized by an infection of the nervous system and is most often transmitted through a bite from an infected animal. For humans, more than 99% of all cases of rabies result from domestic dog bites [3], with approximately 36% of the estimated 59,000 human deaths attributable to rabies globally each

year occurring across continental Africa [2]. In March 2016, the World Health Organization, the World Organisation for Animal Health, the Food and Agriculture Organization of the United Nations, and the Global Alliance for Rabies Control released a global framework for the elimination of dog-mediated human rabies to guide rabies elimination efforts across the globe by 2030. This framework highlights the importance of mass vaccination campaigns of domestic dogs for successful elimination of dog-mediated human rabies.

Many African countries have already embraced mass vaccination of domestic dogs as a reliable elimination strategy for rabies [4–6], resulting in coordinated efforts at the national level in Kenya [6–8], Mozambique [9], Swaziland [10], Tanzania [11, 12], Uganda [13] and Zimbabwe [14]. Over the last decade, a number of additional, coordinated efforts to implement and improve mass dog vaccinations have arisen including large efforts in Chad [15, 16], Malawi [17], and Mali [18]. However, with the exception of Tanzania, which has one of the most comprehensive rural rabies research programs [5], most of these vaccination efforts have tended to focus on urban environments, with densely concentrated human populations [17–20] although a few additional studies have investigated the effectiveness of mass vaccination campaigns in rural African communities [21–23]. Other large scale efforts at eliminating dog-mediated rabies from rural communities have occurred in Asia ([24] and references therein) and Latin America ([25]and references therein) but regional differences in domestic dog ecology and dog ownership practices somewhat limit their transferability to rural communities in Africa. Given the higher risk posed to rural communities by rabies [26] and the additional benefit of rabies control for threatened wildlife [27], a concerted effort to expand mass vaccination efforts into these areas is needed to effectively combat the disease. Although demonstrably feasible [28, 29], vaccination efforts focused in more rural areas often have mixed results when it comes to successful implementation of large-scale campaigns [21, 22, 30, 31].

The success of vaccination campaigns is often measured by the percentage of the domestic dog population that receives vaccination, or coverage rate [32]. This however needs to be complemented by documentation of decreased incidence of the disease through active surveillance over time so as to accurately estimate changes in canine rabies burden in a given area [33]. Campaigns that achieve a sustained immunization of 70% of dogs in a given area [34] are thought to be capable of eliminating rabies from the dog population [35], although observed coverages resulting in successful control of the disease can be lower [32]. In general, the 70% coverage rate is achieved less frequently in rural communities compared to urban centers [36]. This lower coverage can be attributed to a number of factors including accessibility, socio-economic factors, and/or semi-nomadic lifestyles of dog owners [28, 37]. As such, successful implementation of large-scale dog vaccination campaigns often relies upon adaptive-management and site specific approaches [28]. Although a few studies have documented challenges and solutions associated with grassroots vaccination campaigns [21–23], given the region and site-specific success of these campaigns, additional studies from different regions of Africa are warranted.

In Kenya, the elimination of dog-mediated rabies is coordinated by the Zoonotic Disease Unit (ZDU), a collaborative entity consisting of members of the Ministry of Health and the Ministry of Agriculture, Livestock, and Fisheries. The National Rabies Elimination Coordination Committee (NRECC) oversees the implementation and adaptation of the 2014 national strategy for eliminating dog-mediated rabies in the country. The NRECC identified several pilot Counties based on presence or absence of natural barriers such as Lake Victoria (e.g., Kisumu and Siaya Counties) or high numbers of human rabies cases (e.g., Machakos, Kitui, and Makueni Counties) to focus vaccination efforts [8]. With the exception of Kitui County, most of these counties represent densely populated areas dominated by subsistence agriculture located in the southern part of the country [38]. Although guided by the national strategy, the

ZDU has been supportive of grassroots efforts to vaccinate dogs in other communities. Thus, in collaboration with the ZDU and NRECC we sought to examine the success of implementing a grassroots vaccination campaign in Laikipia County, a predominantly pastoral County in central Kenya.

Laikipia County, Kenya, covers 8,696 km$^2$ with an estimated human population of 399,227 (45/km$^2$) and is dominated by livestock ranching and ecotourism in drier northern regions and agro-pastoralism in more mesic southern regions [39]. The remote location and often mobile lifestyle of pastoralists in northern Laikipia presents a unique challenge for mass vaccination currently unaddressed in Kenya. The emergence of the Laikipia Rabies Vaccination Campaign (LRVC) resulted from an increased awareness of the risk of dog-mediated rabies to human health in Laikipia County [40], the role of rabies as a threat to the local population of wild dogs (*Lycaon pictus*), a globally endangered wildlife species [41], and a high human health risk in pastoral communities surrounding Mpala Research Centre (MRC). Researchers at MRC studying the spatial ecology of domestic dogs [42] realized the benefits free vaccinations could provide in terms of improving human, domestic animal, and wildlife health and together with Kenyan veterinarians set about developing the LRVC.

Initiated in 2015, the LRVC continued to grow in scope and scale over the subsequent three years, producing enough data to develop a retrospective assessment of successes and failures. In this paper, we examine the evolution of the LRVC from a small, crowdfunded campaign to a large, multi-organizational vaccination effort, discussing its effectiveness as well as lessons learned. The three-year time frame provides a unique opportunity to examine what aspects of the LRVC are succeeding and those that are not, allowing us to develop an adaptive strategy as we work to increase the scope and scale of the campaign to all of Laikipia County. The insight provided through a detailed assessment of the LRVC should assist other, grassroots campaigns in Africa as well as provide guidance for current and future grassroots campaigns sprouting up in other parts of the world where rabies is prevalent.

## Methods

### Study area

The mass vaccination of domestic dogs against rabies took place in Laikipia County, Kenya, with MRC serving as the central location for operations and logistics (Fig 1, dark gray shaded property). Although land-use practices vary across the County they can be classified into a subset of dominant activities including pastoralism, agro-pastoralism, semi-permanent settlements, and permanent urban centers [43, 44]. Laikipia County has one of the largest concentrations of wildlife in Kenya outside of any National Park [44], creating a unique juxtaposition of privately-owned conservancies and agro-pastoral communities, where humans, wildlife, and domestic animals, including domestic dogs, regularly interact [45, 46]. This diverse landscape differs somewhat from counties targeted for pilot vaccination efforts as part of Kenya's National Rabies Elimination Strategy [8], providing a unique landscape to develop and refine mass-dog vaccination for the less densely populated regions of Kenya, which cover roughly 1/2 of the country.

Communities targeted for vaccinations were selected by proximity to properties focused on wildlife conservation in Laikipia County with known cases of rabies. The communities were selected in the consideration of; (1) availability of financial and logistical support from local partners to conduct dog vaccinations in specific areas, (2) interests by specific community leaders in having vaccinations conducted in their communities, and (3) the need to extend the vaccination area without leaving adjacent communities unvaccinated. This third criterion meant that, in expanding the coverage of the vaccination efforts, we targeted communities adjacent to areas we previously vaccinated rather than adding new, disconnected communities

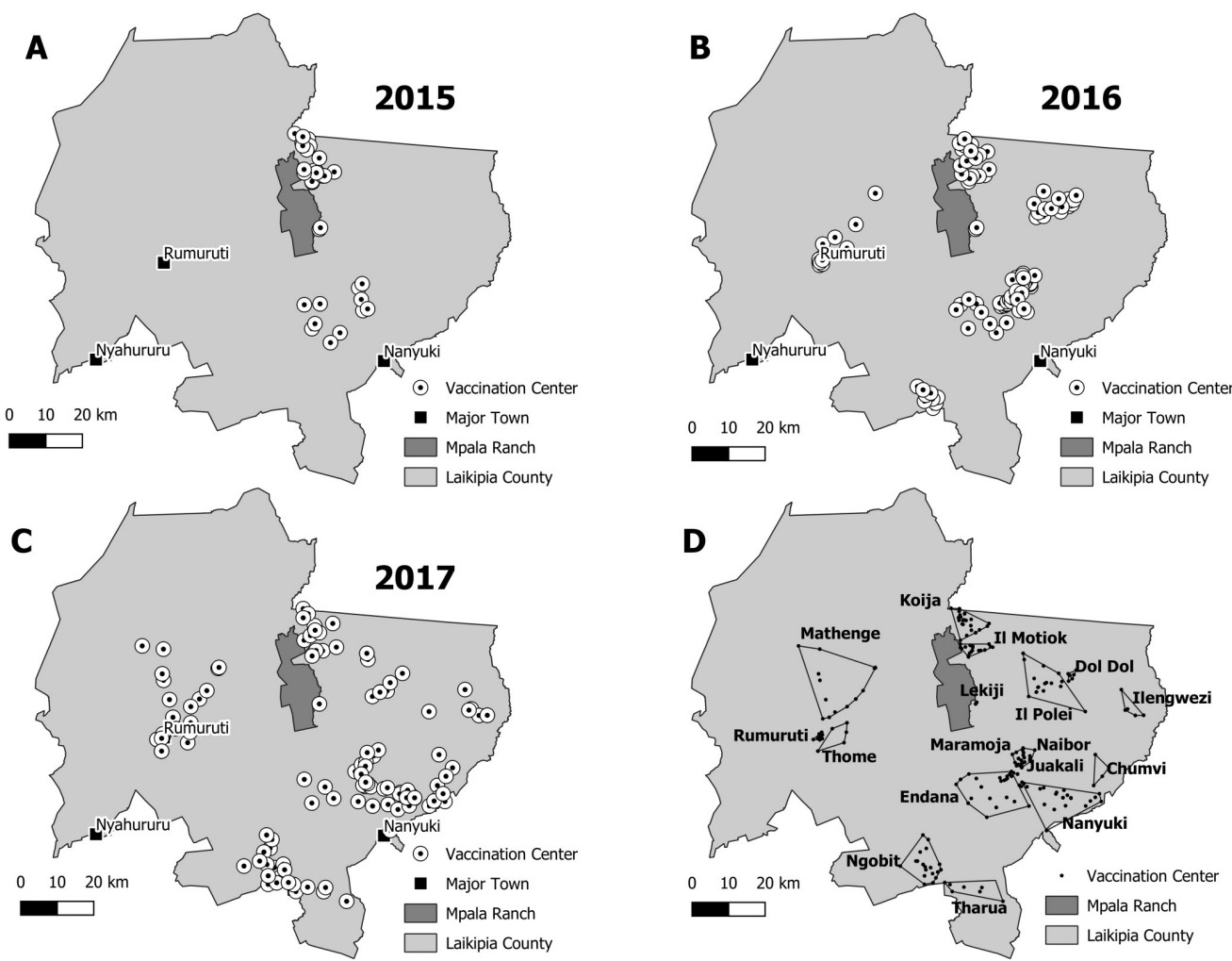

**Fig 1. Central point vaccination stations and community boundaries.** Central point vaccination station locations for the Laikipia Rabies Vaccination Campaign in (A) 2015, (B) 2016, and (C) 2017. (D) Minimum convex hulls delimiting the boundaries of the 17 communities targeted for mass dog vaccinations. Maps were generated using QGIS 2.18.11 Geographic Information System from the Open Source Geospatial Foundation Project. http://qgis. osgeo.org with layers sourced from original GPS coordinates in the field (e.g., central point vaccination stations), the World Agroforestry Centre's Geoscience Lab http://landscapeportal.org (i.e., the Laikipia County property boundaries), and ESRI Data & Maps group http://www.arcgis.com (e.g., cities in Laikipia County and Laikipia County boundary).

in other regions of Laikipia. Central point vaccination stations were positioned based on concentrations of households identified using Google Earth satellite imagery (https://www.google.com/earth/) with subsequent modification through advice from the local leaders and community members.

Initial vaccination efforts (2015) were primarily conducted on group ranches, communally owned tracts of land used to support livestock production that are often adjacent to both privately- and Government-owned properties focused on wildlife conservation [44]. People living in these group ranches typically practice a sedentary or semi-mobile pastoralist lifestyle, predominantly securing their livelihoods through livestock keeping. Subsequent vaccination campaigns (2016 and 2017) expanded coverage to include areas dominated by agro-pastoral communities which rely to a lesser extent on livestock than cultivation. Peri-urban areas were also added in the 2016 and 2017 campaigns to examine how vaccination strategies and effectiveness, as measured through percent coverage, might differ across a multi-use landscape

such as that presented in Laikipia County. We used a combination of these land-use practices and locations of central-point vaccination areas to delimit boundaries of each targeted 'community'. A community in this sense was therefore not necessarily reflective of geographic boundaries associated with named properties but instead a unique area defined by vaccination efforts and the dominant land-use type in that area. We use the term 'community' in this sense throughout the remainder of the paper.

## A volunteer-based approach

The LRVC was founded on a principle of volunteerism. Building upon a One Health approach, professionals from the veterinary, wildlife, and medical fields were solicited to participate in the campaign. In addition, student volunteers were recruited to assist with the implementation of the LRVC. The volunteer-based approach worked as a mechanism to both reduce the cost of the campaign and provide broader impacts to a wider group of people. Over the three years, the campaign benefited from voluntary services by veterinarians from the International Livestock Research Institute, university students from Karatina University and researchers from MRC as well as various other organizations, including both private and non-profit organizations. Many of these organizations also provided local logistical support in the form of in-kind contributions (e.g., providing vehicle(s) for a weekend). Key government partnerships included collaborations with the County Government of Laikipia, especially between the Ministry of Agriculture, Livestock, and Fisheries and the Department of Veterinary Services, as well as the Kenyan National Government's ZDU and NRECC.

Given the large turnover between volunteers across weekends but the nearly constant team size per vaccination campaign, we estimated the minimum number of volunteer hours each team contributed towards vaccinating and traveling in the following ways. Team members spent an average of 12 hours vaccinating with an average team size consisting of 7 volunteers (team leader, 2 veterinarians, 3 students, and 1 security guard) this meant a total of 84 hours were spent by each team per day vaccinating. Seeing as the veterinarians typically traveled from Nairobi to MRC (~ 4 hours one way) and student volunteers from Karatina to MRC (~2 hours) we estimated each team spent 14 hours traveling one way per vaccination weekend. We used these estimates to obtain the total number of volunteer hours vaccinating per campaign by multiplying the total number of teams by the total days vaccinating by the 84 hours/team/day vaccinating. For travel, we assumed two days per weekend spent vaccinating (i.e., round-trip) and multiplied the number of days traveling by the total number of teams by the 14 hours/team/day traveling. We then added these two values to obtain an estimate of total volunteer hours per campaign and divided this value by the total number of dogs vaccinated to obtain volunteer-hours spent/dog vaccinated for each of the three years.

## Vaccination strategy

During 2015–2017, the campaign was carried out under the auspices of the County Government Laikipia and in collaboration with the Kenya Government's ZDU, surrounding conservancies and local leaders. Dogs brought to the central point vaccination stations were registered, vaccinated, marked with a temporary dye for identification and a vaccination certificate was issued to the owner of the dog. Dogs were subcutaneously injected with 1 ml Rabisin rabies vaccine (Merial) or Defensor vaccine (Zoetis). Vaccines were provided at no cost by an anonymous donor in 2015 and from the national bank of vaccines maintained by the ZDU in 2016 and 2017. A brief series of questions was administered to the owners of the dogs across all years, with photographs of each vaccinated animal taken and a new vaccination certificate issued each time.

One week prior to the vaccination campaigns, meetings were held with the local leaders in the targeted communities to strategize about the implementation of the campaign, including (a) determining the locations of central-point vaccination stations and (b) recruiting at least 2 local people from every community for 6 days to help with community outreach and vaccinations. At least three days prior to the vaccination campaign, central point stations were marked using sign boards depicting times and dates of vaccinations as well as the LRVC logo. A public announcement system consisting of a driver and broadcaster with a loud-speaker system was employed to advertise the campaign community-wide two days prior and during the campaign activities themselves. Stations were visited at least twice during each weekend in 2015 and 2016, with alternating AM and PM across days, but stations were not visited multiple times in 2017.

Vaccination teams were created one day prior to the vaccination campaign with each team comprising a team leader, between 2–3 vets, at least 3 volunteer students, one security guard, and when possible one hired community member. The number of teams was dependent on the number of vehicles available since each team used a different vehicle for ease in mobility. Only veterinarians registered to practice in Kenya by the Kenya Veterinary Board were legally permitted to vaccinate dogs although the owners of the dogs would often serve as the primary dog handlers. Students were tasked with (1) taking photographs of the dogs, (2) completing questionnaires (S1 Text) and, (3) filling in and issuing the vaccination cards signed by the veterinarians. Photographs of each animal were taken to help develop a baseline of individually recognizable dogs from natural markings. Each team would visit at least two stations a day. At least 2–3 hours were spent at every station. Although central point vaccination stations served as the primary method of vaccination delivery, one mobile team was added for door-to-door vaccinations in 2016 and 2017.

## Population size estimates and vaccination coverage

Domestic dog population sizes for each community were estimated using three approaches. The first approach used mark-resight surveys (S1 Data) in conjunction with a Lincoln-Peterson Index [47]. These surveys were conducted for 2-days immediately following vaccination events where dogs were marked with temporary livestock markers post-vaccination during the 2016 LRVC. The second approach used average number of dogs/household (S2 Data) as reported by household surveys (S2 Text) conducted during 2016–2017 for 7 communities in conjunction with a broader study of dog demography and rabies knowledge in the study area. Due to logistical constraints, not all communities were surveyed using these methods. For communities without either mark-resight surveys or household surveys, a third approach using the average number of dogs/household for that community type (i.e., pastoral, agro-pastoral, permanent-pastoral, permanent) multiplied by the number of households in that specific community was used to estimate the total number of dogs per community. The number of households per community were defined by drawing a 1-km buffer around the minimum convex hull of all central point vaccination stations in that community in QGIS 2.18 [48] (Fig 1D). Using the High Resolution Settlement Layer (HRSL) developed for Kenya [49] and the community-buffers, the total number of settlements (as delimited by the HRSL layer- see https://code.fb.com/connectivity/open-population-datasets-and-open-challenges/ for details on how settlements are defined) was estimated using the 'Select by Location' tool in QGIS including all settlements that either intersected or were completely within the 1-km buffer layer. The total number of settlements were then multiplied by the average number of dog/household as reported from household surveys to estimate the total dog population per community. For a subset of the communities, we also used Google Earth imagery to manually identify

households to compare to the HRSL results. Households were defined by placing a centroid point within the characteristic fence rows that border human dwellings in this region. Such representation of households would be a conservative estimate as there often can be multiple human dwellings in a single fenced area and fencing strategies can differ across land-use types (e.g., peri-urban versus pastoral communities). Clearly population estimates are dependent on how the number of households per community were estimated, and use of HRSL settlements often differed from manually derived values based on digitizing satellite imagery (S1 Fig and S3 Data).

Percent vaccination coverages were estimated by dividing the total number of dogs vaccinated per community by the total number of dogs estimated for that community. For communities with two estimates of total dog population (e.g., mark-resight and household surveys), the percent coverage was estimated for both values independently.

### Costs of vaccination efforts

Costs of the vaccination campaign were tracked as direct expenses (purchases), in-kind contributions, or volunteer time. Purchases included items in 8 categories: vaccines, supplies, transport, allowances, awareness/community outreach, fuel, food/refreshments, and administration fees. In-kind contributions included items in 6 categories: accommodation and food, fuel, supplies, deficit, vehicles, and vaccines. Costs for vaccinating a dog were estimated both with and without in-kind contributions by taking the total amount spent and dividing it by the total number of dogs vaccinated on a per annum basis. In-kind contributions did not include volunteer hours, which were converted to actual currency estimates by multiplying the total volunteer hours by the average hourly rate for a Kenyan veterinarian technician (http://www.salaryexplorer.com/salary-survey.php?loc=111&loctype=1&job=542&jobtype=3).

### Community outreach and education

In 2015, dog owners who provided verbal informed consent were administered a brief, 8 question questionnaire (S1 Text) aimed at addressing whether or not they had prior knowledge of the disease (Q: Have you ever heard about rabies (Y/N)?) and if their dog had ever been vaccinated (Q: Has your dog ever been vaccinated, if yes, for what?). In 2016, a directed education program was conducted in the schools located in the targeted vaccination area prior to the vaccination. In collaboration with the Northern Kenya Conservation Clubs, and teachers at 12 primary schools hosting the Conservation Clubs, an active lesson plan on rabies was developed and taught in more than 10 communities. School children at each of the 12 Conservation Clubs also developed drawings depicting four panes for an educational poster on rabies as part of a competition whereby the best drawings were selected and used to create posters for distributing to participating communities, primary schools, and medical dispensaries across Laikipia County (S2 and S3 Figs).

Broader efforts at outreach involved engagement of local and national media outlets throughout the campaign. Each year, we invited several media outlets to participate in on-the-ground activities of the LRVC. Participating media partners included: Citizen TV, National Television, K24, Kenya Television Network, Wildlife Direct as well as print media sources.

### Data analysis

All statistical analyses on vaccination results were conducted using data from 2017 for all communities with the exception of Dol Dol (Fig 1), which only had data for 2016. Means and standard deviations for the number of dogs vaccinated, dog population sizes, dog density, and percent coverages were estimated for community types and used in either a one-way analysis

of variance (ANOVA) or Kruskal-Wallis test depending on whether or not the data met the assumptions of normality and homoscedasticity. Assumptions of normality were tested using Q-Q plots and the Shapiro-Wilk test. Assumptions of equal variances were tested using residual plots and Levene's test. If either of these two assumptions were violated, group means were analyzed using the Kruskal-Wallis test. Significance was considered using an alpha value of 0.05. When the assumptions of ANOVA were not violated, results were reported as mean ± standard deviation. This was followed by post hoc pairwise comparisons of means using Tukey tests [50]. When the assumptions were violated and a Kruskal-Wallis test was conducted, results were reported as median plus the interquartile range (IQR). Dog demographic data were analyzed separately using dogs/households as reported and extrapolated from household questionnaires and as a combination of mark-resight estimates and dogs/household. In this context, combining these two estimates meant using mark-resight estimates in lieu of household questionnaire estimates when such data existed for a particular community. Correlations between mean percent coverage and community area, dog density, and density of central point vaccination centers were examined using simple linear regressions. Questionnaire data from 2015 were analyzed using a Chi-Square contingency table to test for a relationship between knowledge of rabies (heard of the disease) and whether or not that owner's dog had been vaccinated. All analyses were performed using the statistical software R [51].

### Ethical consideration

Verbal informed consent was obtained from all dog owners or the person presenting the dog for vaccination prior to administering any vaccine or asking any questions. Household questionnaires and survey protocols were reviewed by the Smithsonian Institution's Internal Review Board (PROTOCOL NUMBER: HS18031) and deemed exempt under paragraph 2 of the Exemption section of Smithsonian Directive 606. Handling procedures for dogs was approved by the Smithsonian Institution's National Museum of Natural History Institutional Animal Care and Use Committee (PROTOCOL NUMBER: 2014–11) and adhered to the United States of America's Animal Welfare Act as implemented and regulated by the United States Department of Agriculture's (USDA) Animal and Plant Health Inspection Service (APHIS).

## Results

### Vaccination coverage area

A total of 17 spatially defined 'communities' received free dog and cat vaccinations against rabies during 2015–2017 as part of the LRVC (Fig 1D). A majority of vaccination efforts were concentrated in the eastern portion of the county, although 2016 and 2017 saw expansion into the southern and western parts of the county (Fig 1A, 1B and 1C). The 17 communities, as classified by livelihood practices and housing structures, were divided into 4 community types: agro-pastoral, pastoral, pastoral/permanent, and permanent settlements (Table 1). In total, approximately 1,500 km$^2$ were covered by vaccination efforts, although community area varied by community type (Table 1).

### Volunteer contributions

A total of 133 unique individuals volunteered at least once across the three years of the LRVC, with more than a quarter participating across multiple years ($n = 36$, 27%). A total of 61 unique veterinarians contributed their time, as well as 45 Karatina University students, and 22 research volunteers in addition to larger numbers of community volunteers, security

**Table 1. Community by community summaries for domestic dog vaccinations against rabies during the Laikipia Rabies Vaccination Campaign, 2015–2017, Laikipia County, Kenya.**

| Community | Type | Pop Estimate¶ | Number of Stations | | | Number of Dogs Vaccinated with Estimates of Percent Coverage | | | | | |
|---|---|---|---|---|---|---|---|---|---|---|---|
| | | | 2015 | 2016 | 2017 | 2015 | % Coverage | 2016 | % Coverage | 2017 | % Coverage |
| **Chumvi** | Agro-Pastoral | NA 977 | - | - | 7 | - | - - | - | - - | 275 | - 28% |
| Endana | Agro-Pastoral | NA 1338 | 6 | 19* | 7 | 177 | - 13% | 423 | - 32% | 471 | - 35% |
| **Nanyuki** | Agro-Pastoral | NA 13,190 | 1 | 2 | 33 | 70 | - 0.5% | 98 | - 0.7% | 2308 | - 17% |
| **Ngobit** | Agro-Pastoral | NA 3548 | - | 10 | 20 | - | - - | 465 | - 13% | 870 | - 25% |
| **Tharua** | Agro-Pastoral | NA 3933 | - | - | 8 | - | - - | - | - - | 488 | - 14% |
| **Thome** | Agro-Pastoral | NA 1329 | - | 1 | 8 | - | - - | 183 | - 14% | 499 | - 38% |
| Il Motiok | Pastoral | 248.9±104.9 442 | 14 | 7 | 5 | 108 | 43% (74–31) 24% | 210 | 84% (100–59) 47% | 88 | 35% (61–25) 20% |
| **Il Polei** | Pastoral | 460.4±89.9 2653 | - | 11 | 12 | - | - - | 444 | 96% (100–81) 17% | 325 | 70% (88–59) 12% |
| **Ilngwezi** | Pastoral | NA 485 | - | - | 9 | - | - - | - | - - | 305 | - 63% |
| Koija | Pastoral | 335.1±76.5 1368 | 13 | 11 | 9 | 130 | 39% (52–26) 9% | 297 | 89% (100–59) 22% | 158 | 47% (65–31) 11% |
| Maramoja | Pastoral | 842.7±457.9 763 | 2 | 25** | 9 | 135 | 16% (35–10) 18% | 540 | 64% (100–41) 71% | 258 | 31% (67–20) 34% |
| **Mathenge** | Pastoral | NA 1913 | - | 4 | 13 | - | - - | 330 | - 17% | 699 | - 37% |
| **Dol Dol** | Pastoral/Permanent | 486.5±176.2 724 | - | 6 | 0 | - | - - | 213 | 43% (69–32) 29% | - | - - |
| Lekiji | Pastoral/Permanent | NA 73 | 2 | 2 | 1 | 37 | - 51% | 55 | - 75% | 39 | - 53% |
| Juakali | Permanent | NA 255 | - | 1 | 2 | 15 | - 6% | 35 | - 14% | 73 | - 29% |
| Naibor | Permanent | 308.8±74.5 299 | 1 | 1 | 2 | 64 | 21% (27–17) 21% | 120 | 39% (51–31) 40% | 109 | 35% (47–28) 36% |
| **Rumuruti** | Permanent | NA 985 | - | 9 | 9 | - | - - | 667 | - 68% | 760 | - 77% |
| Unknown | Unknown | NA NA | 1 | - | 15 | 7 | - - | - | - - | 607 | - - |
| **TOTAL** | | 34,275 | 41 | 65 | 169 | 743 | 2% | 4080 | 12% | 8332 | 24% |

*Includes 9 door-to-door stops

**Includes 14 door-to-door stops

¶Population estimates based on mark-resight (upper) and average dog per household (lower) with corresponding percent coverages for each estimate.

**Boldfaced** community population estimates are based on data on average number of dogs per household gathered from other categorically similar communities.

personnel, and MRC staff. The total number of volunteer hours spent vaccinating increased across the three years and totaled 11,256 hours whereas travel time also increased but totaled far less at 1,876 hours (S3 Text). Dividing the total number of volunteer hours by the total number of dogs vaccinated that year resulted in volunteer hours/dog vaccinated of 1.6 h, 1.2 h, and 0.8 h, in 2015, 2016, and 2017, respectively (S3 Text). Using an average annual salary of 99,000 KES as reported for a Kenyan veterinarian technician (http://www.salaryexplorer.com/

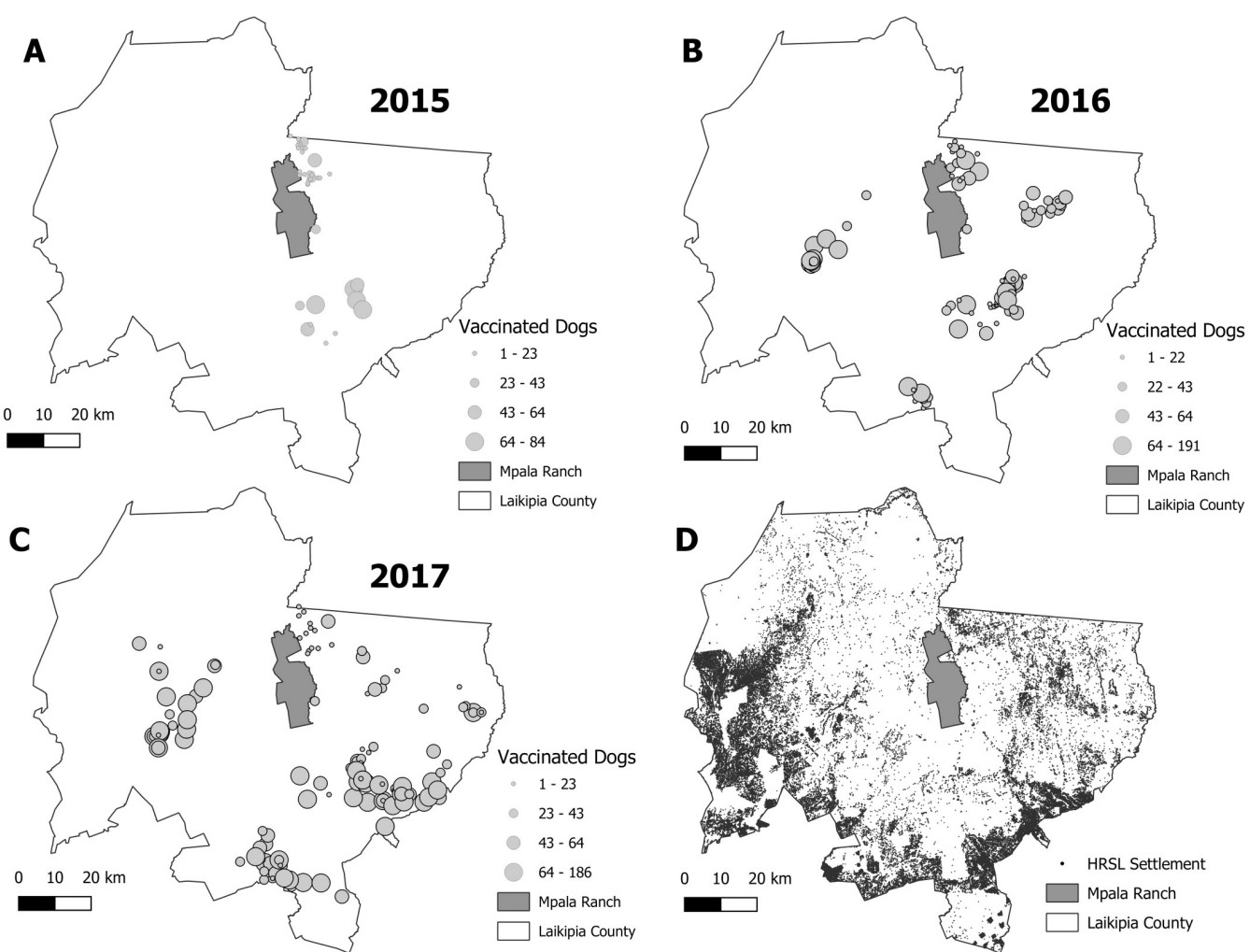

**Fig 2. Total number of dogs vaccinated at central point vaccination stations.** Central point vaccination stations depicting total number of dogs vaccinated per station for (A) 2015, (B) 2016, and (C) 2017. (D) Depiction of settlements using the High Resolution Settlement Layer (HRSL) for Laikipia County, Kenya. Maps were generated using QGIS 2.18.11 Geographic Information System from the Open Source Geospatial Foundation Project http://qgis.osgeo.org with layers sourced from original GPS coordinates in the field (e.g., number of dogs vaccinated/station), the World Agroforestry Centre's Geoscience Lab http://landscapeportal.org (i.e., the Laikipia County property boundaries), ESRI Data & Maps group http://www.arcgis.com (e.g., Laikipia County boundary), and the High Resolution Settlement Layer developed by Facebook Connectivity Lab and Center for International Earth Science Information Network—CIESIN—Columbia University https://www.ciesin.columbia.edu in 2016.

salary-survey.php?loc=111&loctype=1&job=542&jobtype=3), we estimated an hourly rate of 415 KES/hour, or approximately \$4.15 USD/hour. Using this as the base rate for each hour contributed by volunteers, volunteers contributed \$4880 USD in 2015, \$20,335 USD in 2016, and \$29,282 USD in 2017, respectively.

## Vaccination results

A total of 1,040 and 13,155 domestic cats and dogs, respectively, were vaccinated against rabies between 2015 and 2017 (Table 1, S4 Data). Number of domestic dogs vaccinated increased each year from 743 in 2015 to 4,080 and 8,332 in 2016 and 2017, respectively (Table 1, Fig 2). The average number of domestic dogs vaccinated in 2017 did not differ by community type (Kruskal-Wallis $\chi 2$ = 6.534, df = 3, p = 0.09), with a 2017 median and IQR of 494, IQR = 302;

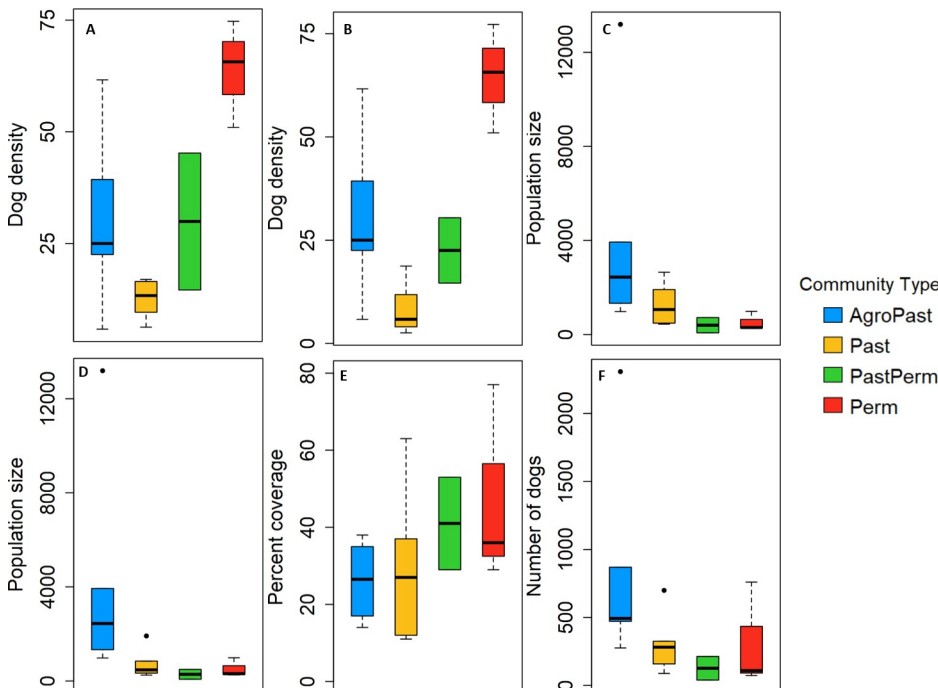

**Fig 3. Domestic dog demographics and percent coverage estimates by community type.** Box plots depicting domestic dog density based on (A) household surveys and (B) combination of mark-resight and household surveys, domestic dog population size using (C) household surveys only and (D) combined household and mark-resight estimates. (E) Percent coverage based on average number of dogs per household estimates and (F) number of dogs vaccinated during 2017.

282, IQR = 137; 126, IQR = 87; and 109, IQR = 344 dogs vaccinated for agro-pastoral, pastoral, pastoral/permanent, and permanent communities, respectively (Fig 3). Efforts and vaccination team structure varied across years (S3 Text), with personnel split into 2 teams for 2015, 6 teams (5 static, 1 mobile) for 2016, and 6 teams (5 static, 1 mobile) for 2017. In addition, a professional and licensed Kenyan physician accompanied the security team to assist with administering first aid, including human post-exposure prophylaxis, to community members. Two additional communities from Laikipia East (i.e., Ngobit and Tharua) were covered in 2017, resulting in vaccinations in all three of Laikipia County's sub-regions.

## Dog population estimates and percent coverage

Population size estimates of domestic dogs using only dogs/household varied among community types (Kruskal-Wallis $\chi^2$ = 8.05, df = 3, p = 0.05, Fig 3), with agro-pastoral communities (2443, IQR = 2506 dogs) having larger population estimates than pastoral (1066, IQR = 1222 dogs), permanent (299, IQR = 365 dogs), and pastoral-permanent (399, IQR = 323 dogs) community types (Table 1, Fig 3). Population size estimates of domestic dogs using estimates of mark-resight for communities that had such surveys (n = 6) together with dogs/household for the remaining communities without mark-resight data (n = 11, S5 Data) also varied among community types (Kruskal-Wallis $\chi^2$ = 8.93, df = 3, p = 0.03, Fig 3), with agro-pastoral communities (2443, IQR = 2506 dogs) having larger population estimates than pastoral (473, IQR = 387 dogs), permanent (309, IQR = 365 dogs), and pastoral-permanent (280, IQR = 207 dogs) community types (Table 1, Fig 3). Comparing population size estimates based on mark-resight surveys versus household questionnaires resulted in estimates with similar values for

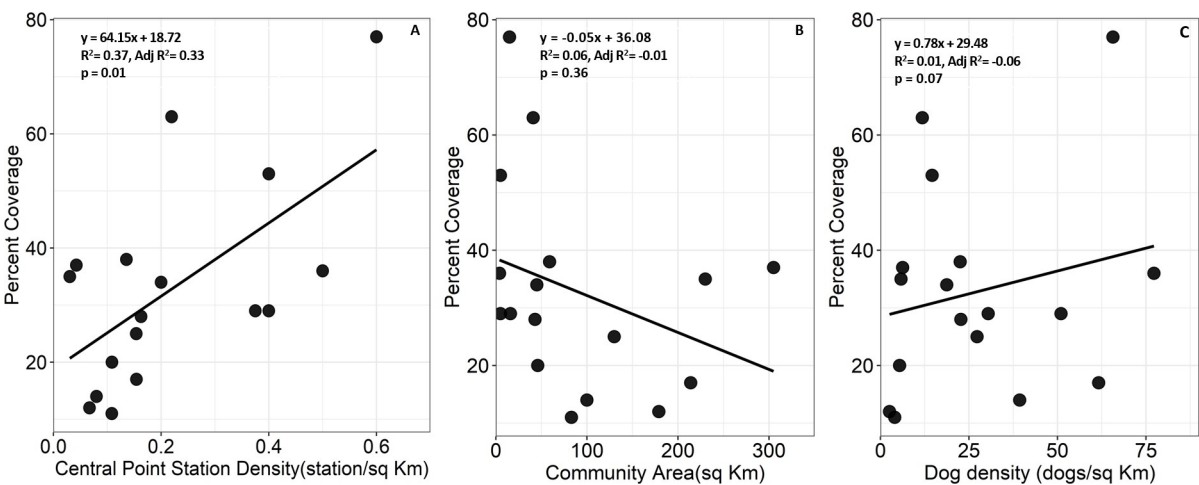

**Fig 4. Predictors of percent coverage estimates.** Simple linear regressions of percent coverage based on (A) household survey estimates of population sizes only for central point station density, (B) community area in km², and (C) dog density per km² for 2017.

some communities (e.g., Dol Dol, Maramoja, and Naibor) and dissimilar values for others (e.g., Il Polei, Il Motiok and Koija). Using the reported average number of dogs per household from the household surveys multiplied by the total number of households contained within the 17 communities, we estimate the total dog population of our vaccination coverage area to be 34,275 animals. The number of dogs totaled 30,709 when using mean population sizes from mark-resight surveys in conjunction with dogs/household estimates, with a lower limit (based on 95% CI of mark-resight estimates) of 29,727 and an upper limit of 31,867 dogs.

Mean population density of dogs based on dogs/household estimates alone varied among community types ($F_{3,13}$ = 8.61, p<0.01, Fig 3) with agro-pastoral (29.9±18.9) and pastoral-permanent areas (29.9±21.7) having similar densities that were higher than those recorded in pastoral (12.7±4.2) areas, but all of which were lower than those in permanent (63.8±12.0) communities. This difference remained significant when using dog density estimates based on a combination of mark-resight and dogs/household values ($F_{3,13}$ = 11.47, p<0.001, Fig 3) with permanent (64.6±13.2) communities maintaining the highest density followed by agro-pastoral (29.9±18.9), pastoral-permanent (22.5±11.2) and pastoral (8.14±6.08). Post hoc tests showed significant differences between permanent communities and the remaining three community types.

Average percent coverage for communities receiving vaccinations in 2017 did not differ significantly among community types ($F_{3,13}$ = 1.02, p = 0.42, Fig 3) with permanent communities having an average of 47.3±25.9% coverage, followed by pastoral-permanent communities with 41.0±17.0%, pastoral communities with 29.5±19.7%, and agro-pastoral communities with 26.2 ±9.5%. Percent coverage appeared to vary significantly with the density of central point vaccination stations ($F_{1,15}$ = 8.99, p = 0.01) but not community area ($F_{1,15}$ = 0.91, p = 0.36) or dog density ($F_{1,15}$ = 0.14, p = 0.71, Fig 4).

Variation in dog population estimates resulting from different techniques affected estimates of percent coverage, with mark-resight estimates on average resulting in lower population size estimates and correspondingly higher percent coverages than those based on dogs/household alone (Table 1). Percent coverage estimates increased or were similar across years with the exception of the pastoral communities of Il Motiok, Il Polei, Koija, and Maramoja, all of which experienced decreased percent coverages between 2016 and 2017 (Table 1). Given a total estimated population of 34,275 dogs for the vaccination coverage area and the total numbers of

**Table 2. Financial summary for the Laikipia Rabies Vaccination Campaign for 2015–2017.**

| | | 2015 | | 2016 | | 2017 | |
|---|---|---|---|---|---|---|---|
| **EXPENSES** | | | | | | | |
| *Purchased* | | KES | USD | KES | USD | KES | USD |
| Vaccines | | 0 | 0 | 177,500 | 1,775 | 0 | 0 |
| Supplies | | 191,400 | 1,914 | 337,425 | 3,374 | 293,595 | 2,936 |
| Transport | | 21,400 | 214 | 161,459 | 1,615 | 369,860 | 3,699 |
| Allowances | | 126,000 | 1,260 | 167,650 | 1,677 | 130,117 | 1,301 |
| Awareness/Comm. Outreach | | 1,600 | 16 | 97,400 | 974 | 109,060 | 1,091 |
| Fuel | | 0 | 0 | 222,290 | 2,223 | 271,084 | 2,711 |
| Food/Refreshments | | 12,870 | 129 | 170,828 | 1,708 | 423,310 | 4,233 |
| Administration fees | | 0 | 0 | 75,000 | 750 | 167,000 | 1,670 |
| | Total | 353,270 | 3,533 | 1,409,552 | 14,096 | 1,764,026 | 17,640 |
| *In kind* | | | | | | | |
| Accommodation & Food* | | 90,000 | 900 | 1,350,000 | 13,500 | 2,160,000 | 21,600 |
| Fuel* | | 23,936 | 239 | 0 | 0 | 0 | 0 |
| Supplies† | | 0 | 0 | 0 | 0 | 450,000 | 4,500 |
| Deficit* | | 0 | 0 | 338,700 | 3,387 | 297,133 | 2,971 |
| Vehicles* | | 20,000 | 200 | 250,000 | 2,500 | 420,000 | 4,200 |
| Vaccines‡ | | 50,000 | 500 | 15,000 | 150 | 612,000 | 6,120 |
| | Total | 183,936 | 1,839 | 1,953,700 | 19,537 | 3,939,133 | 39,391 |
| **GRAND TOTAL** | | **537,206** | **5,372** | **3,363,252** | **33,633** | **5,703,159** | **57,032** |
| **INCOMING** | | | | | | | |
| GoFundMe crowdfunding | | | **5,361** | | **6,130** | | **1,699** |
| Mpala Wildlife Foundation | | | | | **3,000** | | **5,000** |
| Mpala Research Centre | | | | | **1,000** | | **2,000** |
| MRC Board Member | | | | | **1,000** | | |
| Private Donation | | | | | **4,000** | | **3,000** |
| RAW Africa Donation | | | | | | | **1,000** |
| Laikipia Wildlife Forum | | | | | | | **6,000** |
| **GRAND TOTAL** | | | **5,361** | | **15,130** | | **18,699** |

*- In-kind donations covered by Mpala Research Centre (MRC)

†-In-kind donation provided by the County Government of Laikipia's Department of Veterinary Services

‡- In-kind donation provided by the National Government's Zoonotic Disease Unit in consultation with the County Government of Laikipia

dogs vaccinated in 2015, 2016, and 2017 we achieved percent coverage rates of 2%, 12%, and 24%, respectively. Although increasing with each year, only 3 of 38 community-years of vaccination exceeded the 70% target. A community-year of vaccination in this context can be defined as a single community experiencing vaccination in a given year with no implication of duration or consistency of vaccination efforts for that given community.

## Costs of implementing the LRVC

Financial support in the shape of in-kind donations from various partner organizations helped to reduce overall costs per animal (Table 2). Excluding these in-kind donations, the costs of the campaigns were $3,533 USD, $14,096 USD, and $17,640 USD for 2015, 2016, and 2017, respectively. Incorporating in-kind donations as USD equivalents for items including accommodation and food, fuel, supplies, vehicles, and vaccines raised the total costs to $5,372 USD, $33,633 USD, and $57,032 USD for 2015, 2016, and 2017, respectively. Incorporating cost

estimates of volunteer hours contributed as USD equivalents raised the total costs to $10,252 USD, $53,968 USD, and $86,313 USD for 2015, 2016, and 2017, respectively. Direct funds to support the campaign totaled $5,361 USD in 2015, $15,130 USD in 2016, and $18,699 USD in 2017 with the entire budget of 2015 and nearly half of the 2016 budget coming from private donations via crowdfunding on www.gofundme.com (Table 2). Dividing the total costs by the total number of dogs vaccinated resulted in an average cost per animal of $3.44 USD *with* in-kind contributions, $7.44 USD *without* in-kind contributions, and $12.46 USD *without* in-kind contributions and costs saved through volunteer hours. Cost per animal vaccinated based on actual expenses only decreased across all three years, dropping from $4.76 USD in 2015 to $2.11 USD in 2017. Reductions in costs per dog also declined for costs estimates incorporating in-kind contributions and volunteer time. For the two community types with equal effort, i.e., pastoral and agro-pastoral communities (n = 6 each), dividing the 2017 costs with in-kind contributions resulted in a cost per dog of $9.62 USD and $3.59 USD for pastoral and agro-pastoral communities, respectively.

## Community outreach and education

Results from an informal questionnaire administered during the 2015 LRVC indicated a significant relationship between knowledge of rabies and whether or not their dog had been vaccinated ($\chi^2$ = 7.7861, df = 1, p < 0.01), with 69 (11% of respondents) indicating they had both heard of rabies and vaccinated their dog, 394 (64%) had heard of rabies and not vaccinated their dog, 9 (1.5%) had never heard of rabies but vaccinated their dog, and 145 (24%) had never heard of rabies and never vaccinated their dog. Despite this association, 625 unique owners reported an average of 12% of all animals as being previously vaccinated with 74% of these owners expressing knowledge of the disease. In response to the question "Do you know of anyone that has ever been bitten by ANY dog or cat?" 182 of 620 owners (29%) answered Yes with 170 of those 182 (93%) reporting that the person was treated at the hospital in response to the bite. A total of 4 human rabies deaths were reported by participants in the five communities (Endana, Il Motiok, Koija, Lekiji, and Maramoja) where the questionnaire was administered. Community members reported using domestic dogs mostly for guarding the homestead (n = 283, 51%), herding (n = 112, 20%), or a combination of the two (n = 91, 16%) with other uses including companionship (n = 52), hunting (n = 5), and pest control (n = 1). Population control of domestic dogs is not practiced at the County level and no direct examples of control were recorded during the survey although multiple owners independently expressed an interest in controlling dog reproduction.

## Discussion

Over a three-year period (2015–2017) the LRVC grew from a crowdfunded project led by fewer than 10 individuals and 2–3 organizations that vaccinated around 800 dogs, to a project involving more than 90 individuals incorporating more than 15 organizations that succeeded in vaccinating more than 8,000 dogs. Founded on the concept of volunteerism, not a single participant received a stipend for their involvement with the campaign, with the exceptions of community members hired to assist in community mobilization and the medical professional accompanying the team into the field. One of the most important keys to growth of the LRVC was the support provided by the County Government of Laikipia and the Kenyan National Government. By 2017, the County Government of Laikipia was providing supplies and resources in the range of $5,000–6,000 USD as well as logistical and political support to carry out the campaign. Through access to the OIE vaccine bank channels funded through the EU grant to Kenya, the ZDU provided the more than 15,000 vaccines at no cost to the LRVC

during 2016 and 2017. The political momentum associated with this support helped to develop the LRVC from a small, grassroots campaign into a larger, county-wide effort to eliminate dog-mediated rabies from Laikipia. In addition, as the campaign was implemented by a mixture of Kenyan students and veterinarians, with assistance from many local members from communities across Laikipia during actual vaccination drives, an informal but strong network for future engagement was created that Kenyans could readily identify with and take pride in. Although early attempts to engage public health agencies and professionals were unsuccessful, as the campaign progressed, commitments from the County public health officers did occur via in-kind contributions of human vaccines and medical care. Despite the growth in both campaign size and government support, the LRVC fell well-short of achieving the recommended coverage of 70% with only 3 of 38 community-years achieving higher than 70% across the three year period (2015–2017) and an estimated total coverage of only 24% for the entire study area in 2017. While not analyzed directly, consistency in vaccination efforts across communities could impact coverage rates, although from our data it appears that achieving 70% coverage was not more likely if a community was vaccinated consistently across the three years (Table 1). Failure to achieve anywhere near the 70% threshold despite the growing success of the LRVC highlights the dilemma facing grassroots vaccination campaigns; can such efforts actually contribute to the broader ecosystem-wide transmission-stopping aim of mass vaccination campaigns or are they better relegated to raising awareness and vaccinating dogs in small communities to protect those communities directly?

The answer to this question depends entirely on the ability to improve coverage without substantial increases in campaign costs and logistics. Evaluation of three years of data indicated that although the mean number of dogs vaccinated per community type and percent coverages did not differ significantly, there was variation in percent coverage with the lowest values recorded in pastoral communities and highest coverage reported from permanent communities. These results are in line with other studies that have highlighted the challenges facing vaccination campaigns in rural versus urban environments [21–23, 28]. The fact that the density of central point vaccination stations, not dog density or community area, significantly predicted percent coverage (Fig 4) further supports the idea that central point strategies might not be sufficient to achieve 70% coverage in pastoral communities [28]. The highest coverage rate, 77%, was achieved in a permanent community with more than 0.6 central point stations/km$^2$ which was three times the density of central point stations for the pastoral community with the highest percent coverage (Maramoja, 71%). Two potential mechanisms for increasing coverage in our system could be to increase the density of central point stations to at least 0.6 per km$^2$ or to develop a stronger door-to-door strategy in combination with static central points [17].

An additional challenge faced in pastoral communities is reflected in the nearly 50% drop in coverage observed between 2016 and 2017 for 3 of 6 of the pastoral communities, a problem best illustrated with the case of Dol Dol (Fig 1), a small town surrounded by pastoral lands (permanent-pastoral). The Dol Dol example highlights an important lesson on the need to incorporate explicit, adult-based educational outreach in conjunction with vaccination campaigns as well as the risk stochastic factors can have on a campaign's success. Between the 2016 and 2017 LRVC, there was an outbreak of canine distemper virus across Laikipia County, resulting in the deaths of a number of domestic dogs and wild carnivores. Although we managed to successfully vaccinate more than 200 domestic dogs at Dol Dol during 2016, upon returning in 2017 we were met with complete resistance, which community members indicated was a result of the deaths of their dogs, which they attributed to the vaccinations, not the distemper outbreak. This perception of links between the LRVC and dog deaths was also verbally expressed by community members in the three pastoral communities (Il Motiok, Koija, and Maramoja), which experienced 50% declines in vaccination coverage between 2016 and

2017. The hostile attitudes with which the LRVC was met at Dol Dol, and myths associated with the goals and outcomes of vaccination, highlight the need to account for cultural interpretations of such campaigns. Although education efforts were part of the LRVC, these tended to focus on primary school aged children as they are disproportionately affected by rabies [52, 53] and tend to spend substantial amounts of time with domestic dogs. Hosting stakeholder meetings and open dialogue/discussions with entire communities prior to implementing vaccination efforts could go a long way towards dispelling myths and wariness associated with grassroots vaccination efforts, but this certainly adds additional logistical and financial constraints on such campaigns. An alternative approach that has proven successful in other countries is the enforcement of laws that require dog owners to have their animals vaccinated [54]. Although legally required across the country [55], forced implementation of dog vaccinations could be logistically challenging to implement in rural Kenya, especially if dog owners are expected to pay out of pocket for such vaccinations. Given the observed resistance to no-cost vaccinations, revisiting the effectiveness of enforcement of the existing laws certainly poses a philosophical question worth considering.

Although the LRVC was entirely supported by volunteer efforts, such an approach to a large-scale vaccination campaign is not without its shortcomings. Difficulties in sustaining a volunteer-based campaign included restrictions on vaccination follow-up efforts (e.g., household surveys to assess coverage), reduced flexibility in scheduling (e.g., vaccination campaigns were limited to non-working days or weekends), and a lack of consistency in pushing for campaign support (e.g., inability to fund-raise/plan throughout the entire year). Running a volunteer-based program that is largely supported through leadership by personnel with primary responsibilities placed elsewhere (in the case of the LRVC, researchers working on wildlife ecology in the Laikipia County landscape) makes coordinating such a large-scale campaign difficult as the scope of vaccination efforts expands. The large buy-in by multiple conservation, veterinary, and government organizations and diverse volunteer workforce maintained by the LRVC might also represent somewhat of an anomaly given its location within a region characterized by a large number of such organizations, providing ample opportunities for recruiting volunteers, university students, and NGO support. Larger and more remote areas characterized by lower human population densities such as the Kenyan Counties of Turkana or Wajir could present additional challenges for grassroots campaigns seeking to achieve environment-wide elimination of rabies through limited access to such a large and inexpensive volunteer workforce.

As the LRVC progressed, it became abundantly clear that our volunteer-based approach certainly limited our ability to follow up with dog owners to assess critical metrics of success such as coverage rates for communities. The push to vaccinate more and more dogs with less and less post-vaccination monitoring is a serious pitfall we would warn other grassroots campaigns to anticipate and seek to remedy. Such challenges are clearly illustrated by the different estimates of coverage resulting from use of the two population estimates, mark-resight versus household surveys (Table 1). It is imperative that grassroots campaigns discuss explicit ways to gather data for assessing effectiveness of their campaigns, including efforts such as active/passive surveillance systems that go beyond the targeted 70% coverage rate [33]. Active disease surveillance in the Laikipia system was limited during our study but is the target of on-going research efforts and collaborations that involve the LRVC. Such surveillance efforts will yield data vital for addressing the effectiveness of the campaign at eliminating dog-mediated rabies from the County.

Funding is always a limiting factor for implementing successful vaccination efforts at the large scale [28]. Although the first two years of the LRVC were successfully funded via online crowdfunding efforts, as the campaign grew in scale it became clear that such a model was not sustainable. The initial success of the crowdfunding model was contingent upon a large percentage of in-kind donations for food, lodging, and transport by LRVC partners, especially by

MRC. When taking this in-kind support into account, our costs per animal were in line with other rural vaccination efforts (e.g., $2.11 - $4.76 USD). However, in the absence of such in-kind support, the cost per animal, and ultimately the entire campaign, was higher (e.g., $6.84 - $8.24 USD) than what a successful, large-scale campaign should seek to achieve [28]. Adding costs associated with the monetary equivalent of time given by volunteers substantially raises the cost per animal (e.g., $10.14 - $13.40 USD) and helps to provide a numeric value for the benefits of a volunteer-run campaign of this size. Despite being made up of mostly volunteers, our larger team sizes of 6 people required additional in-kind resources for travel and accommodation, adding extra costs that might be reduced through use of smaller, more mobile teams. Although such political buy-in by LRVC partners was critical to its success, it was clear by the third year (Table 2) that the cost of the vaccination campaign was increasing at a rate that would require additional sources of funding beyond in-kind contributions in order to be sustained. However, it does appear that partnering proximal wildlife-focused conservation entities with targeted and adjacent vaccination areas can provide an alternative source of funding and in-kind support for grassroots efforts in other parts of Africa. Such an approach reflects an inverse approach to the One Health concept, whereby funding from organizations primarily interested in wildlife is used to implement a program with major benefits for human health. In order for mass vaccination programs of domestic dogs to be effective at eliminating human rabies, it is worth considering the need for a shift in funding philosophy whereby human health budgets are used to support such vaccination programs. Such a funding model would be more in line with a One Health approach to eliminating dog rabies in human populations and has proven successful in countries such as the Philippines [56]. Intersectoral buy-in, especially between the public health sector and non-public health agencies is critical for successfully developing, implementing, and sustaining effective vaccination campaigns [56–58].

## Conclusions

The LRVC provides an example of the evolution of a grassroots, local vaccination campaign into a large-scale effort to eliminate dog-mediated rabies in rural Kenya, providing a unique opportunity to highlight successes and failures for other future campaigns. The shortfall of achieving the 70% coverage needed to eliminate domestic-dog mediated rabies helped to identify mechanisms that might improve future coverage, which for the LRVC include improved funding that increases the density of central-point stations; implement a stronger door-to-door strategy for pastoral communities; and focuses efforts on post-campaign assessments of coverage based on sound demographic surveys of dog populations. Successful lessons to take away from the LRVC include the value of volunteer-based campaigns; the need to work with local and national governments to both integrate and implement a grassroots campaign in a broader context; and the benefits of involving a diverse group of stakeholders and organizations at both the local, national, and international scale. Potential pitfalls to avoid include lack of development of a central, organized body of stakeholders to guide vaccination efforts and share the workload; failing to define explicit metrics of success and allocating the necessary time and resources to follow-up on vaccination efforts; and lack of incorporating a strong educational, adult-based outreach program to explain the benefits and dispel potential myths associated with vaccination efforts.

## Supporting information

**S1 Data. Mark-resight survey details.** Details of mark-resight surveys conducted for 6 communities during the 2016 Laikipia Rabies Vaccination Campaign
(XLSX)

**S2 Data. Domestic dogs per household.** Summarized results for household demographic surveys implemented in 7 communities targeted for vaccination efforts in Laikipia County, Kenya.
(XLSX)

**S3 Data. Domestic dog population estimates.** Population estimates for all communities based on average number of dogs reported per household multiplied by the number of households.
(XLSX)

**S4 Data. Raw data.** Data on each animal vaccinated in field for 2015, 2016, 2017 used to estimate the number of animals per year, station, and community.
(XLSX)

**S5 Data. Lincoln-Peterson calculations.** Domestic dog population size estimates based on mark-resight surveys conducted in 6 communities during the 2016 Laikipia Rabies Vaccination Campaign.
(XLSX)

**S1 Text. Owner questionnaire.**
(DOCX)

**S2 Text. Household demographic questionnaire.**
(DOCX)

**S3 Text. Personnel and logistic summaries.**
(DOCX)

**S1 Fig. Community maps.** Community by community maps depicting central point station locations, 1 km buffer, and household locations for all 17 communities based on High Resolution Settlement Layer and manually georeferenced households.
(PDF)

**S2 Fig. Educational poster.** Laikipia Rabies Vaccination Campaign educational posters distributed to 12 primary schools as part of the Northern Kenyan Conservation Clubs Program.
(PDF)

**S3 Fig. Informational poster.** Poster used for mobilization of dog owners displayed in communities during Laikipia Rabies Vaccination Campaign.
(PDF)

## Acknowledgments

We would like to thank the following LRVC partner organizations and individuals for their support of the LRVC in a multitude of ways: D. Bahati and M. Kagai of the African Network for Animal Welfare; V. Benka and J. Briggs of the Alliance for Contraception in Cats and Dogs; M. Dyer of the Borana Conservancy; Dr. T. Ndungu, F. M. Muselela, Hon. J. Putunoi, and Dr. L. Murugi of the County Government of Laikipia; B. Marks of the Field Museum of Natural History; J. Akoko of the International Livestock Research Institute; J. M. Mware, and A. Mwangi of Karatina University; M. Mutinda of the Kenya Wildlife Service; H. O'Neill and S. Strebel of the Kenya Rangelands Wild Dog and Cheetah Project; S. Grattan of the Kenya Society for Protection and Care of Animals; P. Hetz, J. Gitonga, and E. Obuchere of the Laikipia Wildlife Forum; A. Takita of the Mara Conservancy; P. Maina of the Ministry of Health;

Dr. D. Thuo of Ministry of Agriculture, Livestock, and Fisheries; M. Littlewood of Mpala Ranch; M. Kinnaird of World Wildlife Foundation; C. Kimsey of the National Science Foundation; National Geographic Society; J. Gayner of Ol Jogi Limited; M. Muthoki of Ol Pejeta Conservancy; B. Squires of RAW Africa; Rufford Foundation; P. Lokeny, D. Mosiany, and J. Exeley of the Small Carnivore Research and Parasite Study; K. Jones, L. Shields, and D. Zimmerman of the Smithsonian Global Health Team, Smithsonian National Zoological Park; B. Schmidt and K. Helgen of the Smithsonian National Museum of Natural History; A. Gardsbane, S. Magda, and B. Miller of Veterinarians International, N. Kahiro of the Zeitz Foundation; and A Bitek, M. Muturi, T. Ndungu, and E. Osoro of the Zoonotic Disease Unit. None of this work would have been possible without the tireless and selfless efforts of the many volunteer veterinarians that participated in the LRVC, in addition to N. Bohr, M. Karani, P. Muinde J. Ngatia, and J. Nyagucha who warrant special recognition, the following veterinarians volunteered for part, and often all, of the LRVC efforts: H. Akoth, P. Akunda, V. Bokonko, V. Chemweno, M. Chepkwony, S. Chirchir, E. Cook, O. O. James, P. Jepkogei, M. Kamau, H. Kameta, N. Kemunto, J. K. Kenana, K. E. Kipchumba, K. B. Kiplann'at, E. Kirwa, V. Kisa, P. N. Kungu, E. N. Kwoba, M. K. Kwoba, P. C. Lugonzo, S. Masika, A. M. Mbithi, M. Melita, K. Momanyi, M. Mugwes, M. Muthoni, J.Mutura, J. Mwadime, J. W. Mwangi, E. Nafula, W. Nakami, C. Njoroge, O. M. Nyangu, J. Ogachi, A. Ogendo, M. Olum, R. N. Omani, E. A. Onsongo, K. Osore, K. Otingah, B. Oundo, T. K. Philip, M. Phyllis, I. Sing'oei, O. Sing'oei, V. K. Toroitich, B. Wako, M. P. Wambui, P. N. Wamuti, D. W. Wanja, and G. Watene. In addition to volunteer veterinarians, a suite of students from Karatina University also volunteered. Student participants included: M. Adama, S. Agatha, C. W. Aggrey, A. R. Aluoch, R. Ang'ila, E. M. Chacha, J. Chege, A. Chemwa, M. W. Esther, B. N. Gatimu, K. W. Janet, H. Kageche, N. E. Kagwi, M. K. Kamau, M. Kendi, J. Kibui, R. D. Kipsambu, F. Kweyu, R. Mbae, J. Meitamei, B. Mohamud, M. L. Muendo, D. Mugambi, D. Munene, R. Muthike, R. Mutunga, B. Mwika, J. M. Ngasike, S. Ngatia, M. V. Njoroge, I. Njoroge, B. Nyabira, L. Nyambura, R. Ang'ila, M. Sharon, K. K. Vincent, E. Wachira, M. E. Wairimu, G. A. Wambui, N. U. Wandili, and T. S. Wanjiku. A number of foreign and Kenyan researchers also volunteered for the LRVC including: V. Benka, I. Driscoll, S. Heisel, J. Hulke, R. Jakopak, M. M. McDonough, K. Ndung'u, H. O'Neill, S. Strebel, G. Titcomb, K. Tomback, K. Wakia, A. Wambua, and S. Weinstein. A special thank you goes out to the three Princeton in Africa Fellows: Da. Martins, A. Padukone, and Z. Sims for 2015, 2016, and 2017, respectively, without whom the LRVC would have never succeeded. The LRVC would not have been possible without extensive support from many Mpala Research Centre staff.

We thank M. Kroll for development of the LRVC logo, and N. Rubenstein of the Northern Kenyan Conservation Clubs and Da. Martin of Princeton University for help with the educational campaign.

## Author Contributions

**Conceptualization:** Adam W. Ferguson, Dishon Muloi, Dedan K. Ngatia, Wangechi Kiongo, Rosie Woodroffe, Eric M. Fèvre, Dino J. Martins.

**Data curation:** Adam W. Ferguson, Dedan K. Ngatia, Wangechi Kiongo.

**Formal analysis:** Adam W. Ferguson, Dishon Muloi.

**Funding acquisition:** Adam W. Ferguson, Paul W. Webala, Mathew Muturi, Samuel M. Thumbi, Lucy Murugi, Eric M. Fèvre, Suzan Murray, Dino J. Martins.

**Investigation:** Adam W. Ferguson, Dedan K. Ngatia, Wangechi Kiongo, Duncan M. Kimuyu, Moses O. Olum.

**Methodology:** Adam W. Ferguson, Dishon Muloi, Dedan K. Ngatia, Wangechi Kiongo, Mathew Muturi, Samuel M. Thumbi, Rosie Woodroffe, Eric M. Fèvre, Suzan Murray, Dino J. Martins.

**Project administration:** Adam W. Ferguson, Dishon Muloi, Dedan K. Ngatia, Wangechi Kiongo, Duncan M. Kimuyu, Moses O. Olum, Dino J. Martins.

**Resources:** Duncan M. Kimuyu, Paul W. Webala, Moses O. Olum, Mathew Muturi, Samuel M. Thumbi, Rosie Woodroffe, Lucy Murugi, Eric M. Fèvre, Suzan Murray, Dino J. Martins.

**Supervision:** Duncan M. Kimuyu, Paul W. Webala, Rosie Woodroffe, Eric M. Fèvre, Suzan Murray, Dino J. Martins.

**Visualization:** Adam W. Ferguson, Dishon Muloi.

**Writing – original draft:** Adam W. Ferguson, Dishon Muloi, Dedan K. Ngatia.

**Writing – review & editing:** Adam W. Ferguson, Dishon Muloi, Dedan K. Ngatia, Duncan M. Kimuyu, Moses O. Olum, Samuel M. Thumbi, Rosie Woodroffe, Eric M. Fèvre, Dino J. Martins.

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
