## [Decision Letter · Decision Letter 0]

8 Dec 2019

Dear Dr Ferguson:

Thank you very much for submitting your manuscript "Volunteer based approach to dog vaccination campaigns to eliminate human rabies: Lessons from Laikipia County, Kenya" (#PNTD-D-19-01492) for review by PLOS Neglected Tropical Diseases. Your manuscript was fully evaluated at the editorial level and by independent peer reviewers. The reviewers appreciated the attention to an important problem, but raised some substantial concerns about the manuscript as it currently stands. These issues must be addressed before we would be willing to consider a revised version of your study. We cannot, of course, promise publication at that time.

We therefore ask you to modify the manuscript according to the review recommendations before we can consider your manuscript for acceptance. Your revisions should address the specific points made by each reviewer. 

When you are ready to resubmit, please be prepared to upload the following:

(1) A letter containing a detailed list of your responses to the review comments and a description of the changes you have made in the manuscript.

(2) Two versions of the manuscript: one with either highlights or tracked changes denoting where the text has been changed (uploaded as a "Revised Article with Changes Highlighted" file); the other a clean version (uploaded as the article file).

(3) If available, a striking still image (a new image if one is available or an existing one from within your manuscript). If your manuscript is accepted for publication, this image may be featured on our website. Images should ideally be high resolution, eye-catching, single panel images; where one is available, please use 'add file' at the time of resubmission and select 'striking image' as the file type. 

Please provide a short caption, including credits, uploaded as a separate "Other" file. If your image is from someone other than yourself, please ensure that the artist has read and agreed to the terms and conditions of the Creative Commons Attribution License at http://journals.plos.org/plosntds/s/content-license (NOTE: we cannot publish copyrighted images). 

(4) If applicable, we encourage you to add a list of accession numbers/ID numbers for genes and proteins mentioned in the text (these should be listed as a paragraph at the end of the manuscript). You can supply accession numbers for any database, so long as the database is publicly accessible and stable. Examples include LocusLink and SwissProt.

(5) To enhance the reproducibility of your results, we recommend that you deposit your laboratory protocols in protocols.io, where a protocol can be assigned its own identifier (DOI) such that it can be cited independently in the future. For instructions see http://journals.plos.org/plosntds/s/submission-guidelines#loc-methods

While revising your submission, please upload your figure files to the Preflight Analysis and Conversion Engine (PACE) digital diagnostic tool, https://pacev2.apexcovantage.com/ PACE helps ensure that figures meet PLOS requirements. To use PACE, you must first register as a user. Then, login and navigate to the UPLOAD tab, where you will find detailed instructions on how to use the tool. If you encounter any issues or have any questions when using PACE, please email us at figures@plos.org.

We hope to receive your revised manuscript by Feb 06 2020 11:59PM. If you anticipate any delay in its return, we ask that you let us know the expected resubmission date by replying to this email.

To submit a revision, go to https://www.editorialmanager.com/pntd/ and log in as an Author. You will see a menu item call Submission Needing Revision. You will find your submission record there. 

Sincerely,

Abdallah M. Samy, PhD

Guest Editor

David Harley

Deputy Editor

I invited 5 reviews for your manuscript; however, i received only three of them (two of the reviewers are now overdue). I decided to go forward with a decision on your manuscript to avoid any further delay on it; our Journal regulations take care from any possible delay in the response to authors. The reviews raised some substantial concerns that should be addressed before considering a revised version of your manuscript. Please respond carefully to all comments raised by the reviewers before submitting your revised version. Thanks too much for choosing PLOS NTDs for your manuscript!

Reviewer's Responses to Questions

**Key Review Criteria Required for Acceptance?**

**Methods**

-Are the objectives of the study clearly articulated with a clear testable hypothesis stated?

-Is the study design appropriate to address the stated objectives?

-Is the population clearly described and appropriate for the hypothesis being tested?

-Is the sample size sufficient to ensure adequate power to address the hypothesis being tested?

-Were correct statistical analysis used to support conclusions?

-Are there concerns about ethical or regulatory requirements being met?

Reviewer #1: This work titled: 'volunteer-based approach to dog vaccination campaigns to eliminate human rabies: lessons from Laikipia County, Kenya' presents an interesting experience vital in the fight against rabies. Particularly, it highlights interesting control options and their challenges around volunteerism, innovative funding methods and mobile human populations. It is therefore well-recommended for publication. In terms of Methods, the responses to the above questions follow:

-Are the objectives of the study clearly articulated with a clear testable hypothesis stated? : YES

-Is the study design appropriate to address the stated objectives?: YES

-Is the population clearly described and appropriate for the hypothesis being tested? : YES

-Is the sample size sufficient to ensure adequate power to address the hypothesis being tested?: YES, THIS IS AN ACCOUNT OF AN INTERVENTION IN WHICH THE SAMPLE SIZE HAD THE NECESSARY POWER TO ADDRESS THE HYPOTHESES

-Were correct statistical analysis used to support conclusions?: YES

-Are there concerns about ethical or regulatory requirements being met?: NONE

Reviewer #2: 1. This commendable work mainly focused on a successful mass dog vaccination that banked on community-based efforts, and materials and human resources contribution from numerous participants and buy ins to the LRVC. 

- Were there other components that contributed to the positive effect other than the grassroots campaign described? 

- There must have been an animal population control?

- Health care workers have also been mobilized even as vaccinators in countless efforts for mass dog vaccination. It is important to describe how were they involved in the LRVC.

- Involvement of other sectors – public health and safety, education, environment, legal affairs, interior and local government, mass media? 

- If yes, then their contributions should also be described as well.

2. In the section on “Selection of communities” on lines 214-219 – please clarify what the third criteria means. It states “extending the vaccine area without leaving the adjacent communities unvaccinated’

3. Suggest that authors clarify or give an operational definition of “community years of vaccination”. What does this actually mean if vaccination campaigns do not run every month of the year in every community? Not the same number of days or weeks or months across the different communities? This likely affected the vaccination coverage in the different communities across the 3 years of the LRVC

4. In the Community outreach and education section (from line 317) – there was no description of the trimedia or mass media. Was this really so, not even leaflets, posters, radio or TV speils?

5. There were a lot of qualitative data, and health economics data (including direct and indirect costs of a mass vaccination) that have been generated in this study but no clear description of how these were managed and included in the analysis. The data analysis section (lines 332-348) only described statistical analysis of the dog vaccination, dog demographics and household knowledge & practice.

Reviewer #3: The study objectives are clear and the design is appropriate. The population is well-described. My only methodical concern is in the dog population estimates using the mark-resight method. While it is commonly used this way with the day 1 number being the number vaccinated, it technically violates some assumptions of the calculation. The alternative method using household surveys compensates for instability in the mark-resight method. I would appreciate more details on how the two were combined, but that would be better relegated to supplementary files due to the scope of this paper.

**Results**

-Does the analysis presented match the analysis plan?

-Are the results clearly and completely presented?

-Are the figures (Tables, Images) of sufficient quality for clarity?

Reviewer #1: -Does the analysis presented match the analysis plan?: YES

-Are the results clearly and completely presented?: YES

-Are the figures (Tables, Images) of sufficient quality for clarity?: YES

Reviewer #2: 1. A majority of the results section included only data on vaccination coverage, vaccination results, dog demographics, very little data on the input resources like manpower, and enabling factors such as governance, positive externalities etc

2. The implementation costs included only materials and financial contributions. In the methods section, it was stated that volunteerism was used as a strategy to reduce cost of the campaign and have a broader impact to a wider group of people but there was very limited data and information, if at all, on the volunteer manpower. Suggest to include calculations on human resources during the campaigns. An example is the manhours per 100 dogs vaccinated. Indicators like these are important in planning and strategizing campaigns 

3. Monetizing or giving a monetary equivalent to time given by volunteers to the vaccination and education efforts is also an important aspect to analyze, as this may give a better perspective of societal and individual costs expended in mass vaccination campaigns.

Reviewer #3: The analysis is appropriate and the results are clear. The figures and tables could use minor adjustments to improve legibility.

**Conclusions**

-Are the conclusions supported by the data presented?

-Are the limitations of analysis clearly described?

-Do the authors discuss how these data can be helpful to advance our understanding of the topic under study?

-Is public health relevance addressed?

Reviewer #1: -Are the conclusions supported by the data presented?: YES

-Are the limitations of analysis clearly described?: YES

-Do the authors discuss how these data can be helpful to advance our understanding of the topic under study?: YES, ALL DATA ARE PROVIDED

-Is public health relevance addressed?: YES

Reviewer #2: 1. The effort to increase vaccination coverage in dogs and other target animals may have been successful but it did not clearly translate to reducing or eliminating disease transmission since there was no epidemiologic trend nor surveillance data for human and animal rabies presented. Was there any available disease surveillance data both for human and animal rabies?

2. The funding philosophy of shifting human health budgets to support mass dog vaccinations have been adopted in countries in Asia like Thailand and the Philippines. Therefore the authors conclusion only affirms this.

Reviewer #3: The data support the conclusions and the limitations are well-described. The discussion is nuanced and thorough. The authors provide helpful details and lessons for other programs in the planning stages for rabies control.

**Editorial and Data Presentation Modifications?**

Reviewer #1: Recommend to Accept this work with these very minor corrections:

Page 20, Line 421 it should read 'area' not 'are'

Page 27, Line 547, A reference needs to be added at the end of this sentence: "An alternative approach that has proven successful in other countries (reference)..."

Page 29, Line 595, should be 'domestic dogs' and not just domestic

Reviewer #2: (No Response)

Reviewer #3: Author summary: review for typos/punctuation

Introduction: 

Second paragraph line 132, please elaborate on your thought here rather than pointing to the references. In the next sentence a reference (Woodroffe 1999) is missing from the citations.

Third paragraph line 143, those two clauses seem redundant- the first should be removed.

Methods:

First paragraph line 192, specify MRC is dark grey as the whole map is grey.

Line 197 move comma from after to before “regularly”

How were the vaccinations tracked for the campaign managers? What was the purpose of the vaccination photographs?

Results

You can leave out the F-statistic and chi square numbers, they just add clutter.

First sentence comma not necessary

Table 1: alternate line shading would be helpful to read the rows

Figure 3: Y axis labels would help, along with putting C and D on the same row for easy comparison. For E, which population estimate type was used as the denominator?

Last sentence: Were those 5 deaths reported by hospitals or identified from the survey?

Discussion

Second paragraph line 523, Maramoja is said to have 71% coverage but it doesn’t look like there’s another community above 70% in the regression figure. Was that in a different year?

Last paragraph line 595, missing the word “animals” after “domestic”

Conclusions

One aspect not discussed that could impact cost-effectiveness is team size. The teams sound quite large to be handling an animal at a time.

**Summary and General Comments**

Reviewer #1: This is a very well-written research. it highlights important aspects and challenges of rabies controls especially among remotely located and mobile communities, which have some of the highest dependence on domestic dogs. Information on challenges and opportunities of volunteerism and innovative sources of funding will prove useful to the audience of this work.

Reviewer #2: It is striking that there was only one reference from Bhutan cited and no comparisons with dog-mediated rabies prevention, control and elimination programs in Asia. Numerous similar work have been published from Asia with the goal of building a sustainable program that would prevent human rabies by eliminating rabies at its animal source. It would be worthwhile for the authors to review and include citations to these Asian references.

Reviewer #3: This study summarizes three years of work in Laikipia County, Kenya to build a rabies vaccination campaign based on volunteer efforts. The campaign planning and evaluations have been well-described in a way that is comparable to many other campaigns. The authors provide an honest look into the challenges and expenses of reaching rural and non-permanent settlements, while detailing how a small program might offset those costs by advocating for donations and coordinating volunteers.

PLOS authors have the option to publish the peer review history of their article (what does this mean?). If published, this will include your full peer review and any attached files.

Reviewer #1: Yes: Dr. Emmanuel Abraham Mpolya

Reviewer #2: No

Reviewer #3: No

---

## [Decision Letter · Decision Letter 1]

29 Mar 2020

Dear Dr Ferguson,

We are pleased to inform you that your manuscript 'Volunteer based approach to dog vaccination campaigns to eliminate human rabies: Lessons from Laikipia County, Kenya' has been provisionally accepted for publication in PLOS Neglected Tropical Diseases.

Best regards,

Abdallah M. Samy, PhD

Deputy Editor

David Harley

Deputy Editor

---

## [Editor Report · Acceptance letter]

18 May 2020

Dear Dr Ferguson,

We are delighted to inform you that your manuscript, "Volunteer based approach to dog vaccination campaigns to eliminate human rabies: Lessons from Laikipia County, Kenya," has been formally accepted for publication in PLOS Neglected Tropical Diseases.

Best regards,

Serap Aksoy

Editor-in-Chief

Shaden Kamhawi

Editor-in-Chief
